# INTRIGUING PROPERTIES OF ADVERSARIAL TRAINING AT SCALE

**Cihang Xie**
Johns Hopkins University

**Alan Yuille**
Johns Hopkins University

## ABSTRACT

Adversarial training is one of the main defenses against adversarial attacks. In this paper, we provide the first rigorous study on diagnosing elements of large-scale adversarial training on ImageNet, which reveals two intriguing properties.

First, we study the role of normalization. Batch Normalization (BN) is a crucial element for achieving state-of-the-art performance on many vision tasks, but we show it may prevent networks from obtaining strong robustness in adversarial training. One unexpected observation is that, for models trained with BN, simply removing clean images from training data largely boosts adversarial robustness, *i.e.*, 18.3%. We relate this phenomenon to the hypothesis that clean images and adversarial images are drawn from two different domains. This two-domain hypothesis may explain the issue of BN when training with a mixture of clean and adversarial images, as estimating normalization statistics of this mixture distribution is challenging. Guided by this two-domain hypothesis, we show disentangling the mixture distribution for normalization, *i.e.*, applying separate BNs to clean and adversarial images for statistics estimation, achieves much stronger robustness. Additionally, we find that enforcing BNs to behave consistently at training and testing can further enhance robustness.

Second, we study the role of network capacity. We find our so-called "deep" networks are still shallow for the task of adversarial learning. Unlike traditional classification tasks where accuracy is only marginally improved by adding more layers to "deep" networks (*e.g.*, ResNet-152), adversarial training exhibits a much stronger demand on deeper networks to achieve higher adversarial robustness. This robustness improvement can be observed substantially and consistently even by pushing the network capacity to an unprecedented scale, *i.e.*, ResNet-638.

## 1 INTRODUCTION

Adversarial attacks (Szegedy et al., 2014) can mislead neural networks to make wrong predictions by adding human imperceptible perturbations to input data. Adversarial training (Goodfellow et al., 2015) is shown to be an effective method to defend against such attacks, which trains neural networks on adversarial images that are generated on-the-fly during training. Later works further improve robustness of adversarially trained models by mitigating gradient masking (Tramèr et al., 2018), imposing logits pairing (Kannan et al., 2018), denoising at feature space (Xie et al., 2019b), *etc*. However, these works mainly focus on justifying the effectiveness of proposed strategies and apply inconsistent pipelines for adversarial training, which leaves revealing important elements for training robust models still a missing piece in current adversarial research.

In this paper, we provide the first rigorous diagnosis of different adversarial learning strategies, under a unified training and testing framework, on the large-scale ImageNet dataset (Russakovsky et al., 2015). We discover two intriguing properties of adversarial training, which are essential for training models with stronger robustness. First, though Batch Normalization (BN) (Ioffe & Szegedy, 2015) is known as a crucial component for achieving state-of-the-arts on many vision tasks, it may become a major obstacle for securing robustness against strong attacks in the context of adversarial training. By training such networks adversarially with different strategies, *e.g.*, imposing logits pairing (Kannan et al., 2018), we observe an unexpected phenomenon — removing clean images from training data is the most effective way for boosting model robustness. We relate this phenomenon

to the conjecture that clean images and adversarial images are drawn from two different domains. This two-domain hypothesis may explain the limitation of BN when training with a mixture of clean and adversarial images, as estimating normalization statistics on this mixture distribution is challenging. We further show that adversarial training without removing clean images can also obtain strong robustness, if the mixture distribution is well disentangled at BN by constructing different mini-batches for clean images and adversarial images to estimate normalization statistics, *i.e.*, one set of BNs exclusively for adversarial images and another set of BNs exclusively for clean images. An alternative solution to avoiding mixture distribution for normalization is to simply replace all BNs with batch-unrelated normalization layers, *e.g.*, group normalization (Wu & He, 2018), where normalization statistics are estimated on each image independently. These facts indicate that model robustness is highly related to normalization in adversarial training. Furthermore, additional performance gain is observed via enforcing consistent behavior of BN during training and testing.

Second, we find that our so-called "deep" networks (*e.g.*, ResNet-152) are still shallow for the task of adversarial learning, and simply going deeper can effectively boost model robustness. Experiments show that directly adding more layers to "deep" networks only marginally improves accuracy for traditional image classification tasks. In contrast, substantial and consistent robustness improvement is witnessed even by pushing the network capacity to an unprecedented scale, *i.e.*, ResNet-638. This phenomenon suggests that larger networks are encouraged for the task of adversarial learning, as the learning target, *i.e.*, adversarial images, is a more complex distribution than clean images to fit.

In summary, our paper reveals two intriguing properties of adversarial training: (1) properly handling normalization is essential for obtaining models with strong robustness; and (2) our so-called "deep" networks are still shallow for the task of adversarial learning. We hope these findings can benefit future research on understanding adversarial training and improving adversarial robustness.

## 2    RELATED WORK

**Adversarial training.**    Adversarial training constitutes the current foundation of state-of-the-arts for defending against adversarial attacks. It is first developed in Goodfellow et al. (2015) where both clean images and adversarial images are used for training. Kannan et al. (2018) propose to improve robustness further by encouraging the logits from the pairs of clean images and adversarial counterparts to be similar. Instead of using both clean and adversarial images for training, Madry et al. (2018) formulate adversarial training as a min-max optimization and train models exclusively on adversarial images. Subsequent works are then proposed to further improve the model robustness (Xie et al., 2019b; Zhang et al., 2019b; Hendrycks et al., 2019a; Qin et al., 2019; Zhang & Wang, 2019; Carmon et al., 2019; Hendrycks et al., 2019b; Alayrac et al., 2019; Zhai et al., 2019) or accelerate the adversarial training process (Shafahi et al., 2019; Zhang et al., 2019a; Wang & Zhang, 2019). However, as these works mainly focus on demonstrating the effectiveness of their proposed mechanisms, a fair and detailed diagnosis of large-scale adversarial training strategies remains as a missing piece. In this work, we provide the first detailed diagnosis which reveals two intriguing properties of training adversarial defenders at scale.

**Normalization Layers.**    Normalization is an effective technique to accelerate the training of deep networks. Different methods are proposed to exploit batch-wise (*e.g.*, BN (Ioffe & Szegedy, 2015)), layer-wise (*e.g.*, layer normalization (Ba et al., 2016)) or channel-wise (*e.g.*, instance normalization (Ulyanov et al., 2016) and group normalization (Wu & He, 2018)) information for estimating normalization statistics. Different from traditional vision tasks where BN usually yields stronger performance than other normalization methods, we show that BN may become a major obstacle for achieving strong robustness in the context of adversarial training, and properly handling normalization is an essential factor to improving adversarial robustness.

## 3    ADVERSARIAL TRAINING FRAMEWORK

As inconsistent adversarial training pipelines were applied in previous works (Kannan et al., 2018; Xie et al., 2019b), it is hard to identify which elements are important for obtaining robust models. To this end, we provide a unified framework to train and to evaluate different models, for the sake of fair comparison.

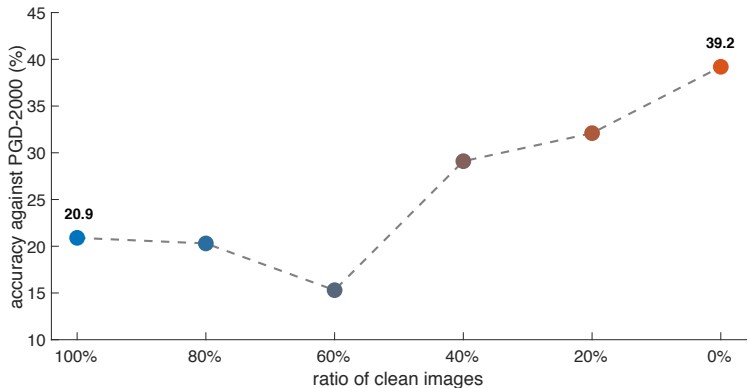

Figure 1: **The relationship between model robustness and the portion of clean images used for training**. We observe that the strongest robustness can be obtained by training completely without clean images, surpassing the baseline model by 18.3% accuracy against PGD-2000 attacker.

**Training Parameters.** We use the publicly available adversarial training pipeline[1] to train *all* models with different strategies on ImageNet. We select ResNet-152 (He et al., 2016) as the baseline network, and apply projected gradient descent (PGD) (Madry et al., 2018) as the adversarial attacker to generate adversarial examples during training. The hyper-parameters of the PGD attacker are: maximum perturbation of each pixel $\epsilon = 16$, attack step size $\alpha = 1$, number of attack iterations N = 30, and the targeted class is selected uniformly at random over the 1000 ImageNet categories. We initialize the adversarial image by the clean counterpart with probability = 0.2, or randomly within the allowed $\epsilon$ cube with probability = 0.8. All models are trained for a total of 110 epochs, and we decrease the learning rate by $10\times$ at the 35-th, 70-th, and 95-th epoch.

**Evaluation.** For performance evaluation, we mainly study *adversarial robustness* (rather than clean image accuracy) in this paper. Specifically, we follow the setting in Kannan et al. (2018) and Xie et al. (2019b), where the targeted PGD attacker is chosen as the white-box attacker to evaluate robustness. The targeted class is selected uniformly at random. We constrain the maximum perturbation of each pixel $\epsilon = 16$, set the attack step size $\alpha = 1$, and measure the robustness by defending against PGD attacker of 2000 attack iterations (*i.e.*, PGD-2000). As in Kannan et al. (2018) and Xie et al. (2019b), we always initialize the adversarial perturbation from a random point within the allowed $\epsilon$-cube.

We apply these training and evaluation settings by default for all experiments, unless otherwise stated.

## 4 EXPLORING NORMALIZATION TECHNIQUES IN ADVERSARIAL TRAINING

### 4.1 ON THE EFFECTS OF CLEAN IMAGES IN ADVERSARIAL TRAINING

In this part, we first elaborate on the effectiveness of different adversarial training strategies on model robustness. Adversarial training can be dated back to Goodfellow et al. (2015), where they mix clean images and the corresponding adversarial counterparts into each mini-batch for training. We choose this strategy as our starting point, and the corresponding loss function is:

$$\hat{J}(\theta, x, y) = \alpha J(\theta, x^{clean}, y) + (1 - \alpha)J(\theta, x^{adv}, y), \qquad (1)$$

where $J(\cdot)$ is the loss function, $\theta$ is the network parameter, $y$ is the ground-truth, and training pairs $\{x^{clean}, x^{adv}\}$ are comprised of clean images and their adversarial counterparts, respectively. The parameter $\alpha$ balances the relative importance between clean image loss and adversarial image loss. We set $\alpha = 0.5$ following Goodfellow et al. (2015). With our adversarial training framework, this model can achieve 20.9% accuracy against PGD-2000 attacker. Besides this baseline, we also study the effectiveness of two recently proposed adversarial training strategies (Madry et al., 2018; Kannan et al., 2018), and provide the results as follows.

---

[1]https://github.com/facebookresearch/ImageNet-Adversarial-Training

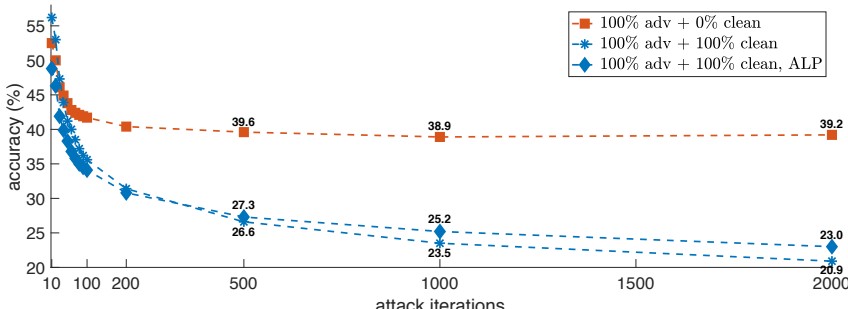

Figure 2: **Comprehensive robust evaluation on ImageNet**. For models trained with different strategies, we show their accuracy against PGD attackers with 10 to 2000 iterations. Only the curve of *100% adv + 0% clean* becomes asymptotic when evaluating against attackers with more iterations.

**Ratio of clean images.** Different from the canonical form in Goodfellow et al. (2015), Madry et al. (2018) apply the min-max formulation for adversarial training where no clean images are used. We note this min-max type optimization can be dated as early as Wald (1945). We hereby investigate the relationship between model robustness and the ratio of clean images used for training. Specifically, for each training mini-batch, we keep adversarial images unchanged, but removing their clean counterparts by 20%, 40%, 60%, 80% and 100%. We report the results in Figure 1. Interestingly, removing a portion of clean images from training data can significantly improve model robustness, and the strongest robustness can be obtained by completely removing clean images from the training set, *i.e.*, it achieves an accuracy of 39.2% against PGD-2000 attacker, outperforming the baseline model by a large margin of 18.3%.

**Adversarial logits pairing.** For performance comparison, we also explore the effectiveness of an alternative training strategy, adversarial logits pairing (ALP) (Kannan et al., 2018). Compared with the canonical form in Goodfellow et al. (2015), ALP imposes an additional loss to encourage the logits from the pairs of clean images and adversarial counterparts to be similar. As shown in Figure 2, our re-implemented ALP obtains an accuracy of 23.0% against PGD-2000 attacker[2], which outperforms the baseline model by 2.1%. Compared with the strategy of removing clean images, this improvement is much smaller.

**Discussion.** Given the results above, we conclude that training exclusively on adversarial images is the most effective strategy for boosting model robustness. For example, by defending against PGD-2000 attacker, the baseline strategy in Goodfellow et al. (2015) (referred to as *100% adv + 100% clean*) obtains an accuracy of 20.9%. Adding a loss of logits pairing (Kannan et al., 2018) (referred to as *100% adv + 100% clean, ALP*) slightly improves the performance by 2.1%, while completely removing clean images (Madry et al., 2018; Xie et al., 2019b) (referred to as *100% adv + 0% clean*) boosts the accuracy by 18.3%. We further plot a comprehensive evaluation curve of these three training strategies in Figure 2, by varying the number of PGD attack iteration from 10 to 2000. Surprisingly, only *100% adv + 0% clean* can ensure model robustness against strong attacks, *i.e.*, performance becomes asymptotic when allowing PGD attacker to perform more attack iterations. Training strategies which involve clean images for training are suspicious to result in worse robustness, if PGD attackers are allowed to perform more attack iterations. In the next section, we will study how to make these training strategies, *i.e.*, *100% adv + 100% clean* and *100% adv + 100% clean, ALP* to secure their robustness against strong attacks.

## 4.2 THE DEVIL IS IN THE BATCH NORMALIZATION

**Two-domain hypothesis.** Compared to feature maps of clean images, Xie et al. (2019b) show that feature maps of their adversarial counterparts tend to be more noisy. Meanwhile, several works (Li

---

[2]Surprisingly, we note our reproduced ALP result is significantly stronger than the result reported in the original ALP paper (Kannan et al., 2018), as well in an independent study (Engstrom et al., 2018). We identify this performance gap is mainly due to different settings of training parameter, and provide a detailed diagnosis in the supplementary material.

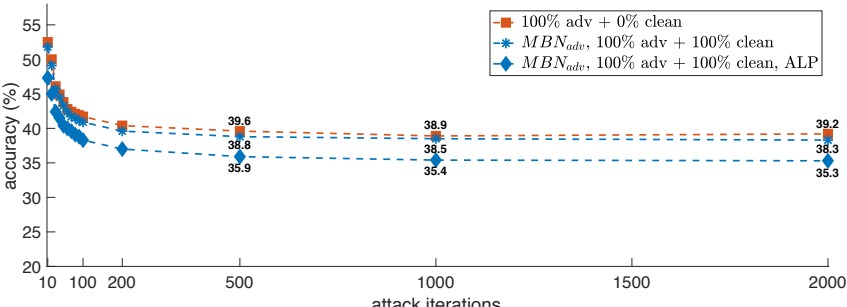

Figure 3: **Disentangling the mixture distribution for normalization secures model robustness**. Unlike the blue curves in Figure 2, these new curves become asymptotic when evaluating against attackers with more iterations, which indicate that the networks using MBN_adv can behave robustly against PGD attackers with different attack iterations, even if clean images are used for training.

& Li, 2017; Metzen et al., 2018; Feinman et al., 2017; Pang et al., 2018; Li et al., 2019) demonstrate it is possible to build classifiers to separate adversarial images from clean images. These studies suggest that *clean images and adversarial images are drawn from two different domains*[3]. This two-domain hypothesis may provide an explanation to the unexpected observation (see Sec. 4.1) and we ask — why simply removing clean images from training data can largely boost adversarial robustness?

As a crucial element for achieving state-of-the-arts on various vision tasks, BN (Ioffe & Szegedy, 2015) is widely adopted in many network architectures, *e.g.*, Inception (Szegedy et al., 2015), ResNet (He et al., 2016) and DenseNet (Huang et al., 2017). The normalization statistics of BN are estimated across different images. However, exploiting batch-wise statistics is a challenging task if input images are drawn from different domains and therefore networks fail to learn a unified representation on this mixture distribution. Given our two-domain hypothesis, when training with both clean and adversarial images, the usage of BN can be the key issue for resulting in weak adversarial robustness in Figure 2.

Based on the analysis above, an intuitive solution arise: *accurately estimating normalization statistics should enable models to train robustly even if clean images and adversarial images are mixed at each training mini-batch*. To this end, we explore two ways, where the mixture distribution is disentangled at normalization layers, for validating this argument: (1) maintaining separate BNs for clean/adversarial images; or (2) replacing BNs with batch-unrelated normalization layers.

**Training with Mixture BN.** Current network architectures estimate BN statistics using the mixed features from both clean and adversarial images, which leads to weak model robustness as shown in Figure 2. Xie et al. (2019a) propose that properly decoupling the normalization statistics for adversarial training can effectively boost image recognition. Here, to study model robustness, we apply Mixture BN (MBN) (Xie et al., 2019a), which disentangles the mixed distribution via constructing different mini-batches for clean and adversarial images for accurate BN statistics estimation (illustrated in Figure 4), *i.e.*, one set of BNs exclusively for adversarial images (referred to as MBN_adv), and another set

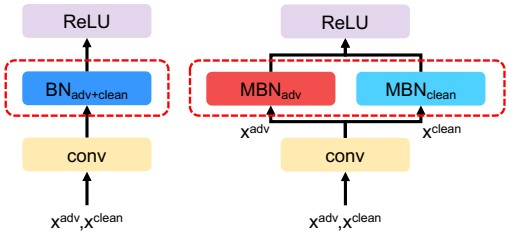

Figure 4: Standard BN (left) estimates normalization statistics on the mixture distribution. MBN (right) disentangles the distribution by constructing different mini-batch for clean and adversarial images to estimate normalization statistics.

of BNs exclusively for clean images (referred to as MBN_clean). We do not change the structure of other layers. We verify the effectiveness of this new architecture with two (previously less robust) training strategies, *i.e.*, *100% adv + 100% clean* and *100% adv + 100% clean, ALP*.

---

[3]Or more precisely, "natural" images collected in the datasets and the corresponding adversarial images may come from two different distributions.

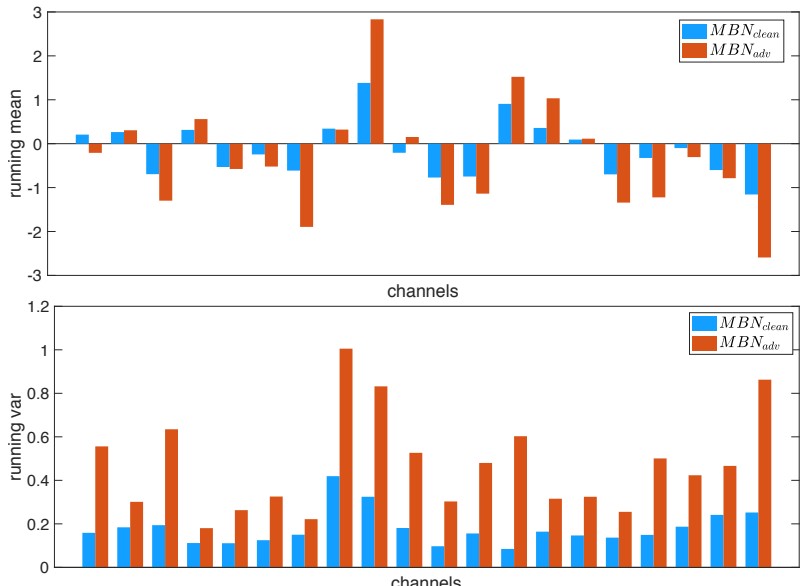

Figure 5: Statistics of running mean and running variance of MBN on randomly sampled 20 channels in a ResNet-152's res₃ block. This suggests that clean and adversarial images induce significantly different normalization statistics.

At inference time, whether an image is adversarial or clean is unknown. We thereby measure the performance of networks by applying either $MBN_{adv}$ or $MBN_{clean}$ separately. The results are shown in Table 1. We find the performance is strongly related to how BN is trained: *when using $MBN_{clean}$, the trained network achieves nearly the same clean image accuracy as the whole network trained exclusively on clean images; when using $MBN_{adv}$, the trained network achieves nearly the same adversarial robustness as the whole network trained exclusively on adversarial images*. Other factors, like whether ALP is applied for training, only cause subtle differences in performance. We further plot an extensive robustness evaluation curve of different training strategies in Figure 3. Unlike Figure 2, we observe that networks using $MBN_{adv}$ now can secure their robustness against strong attacks, *e.g.*, the robustness is asymptotic when increasing attack iterations from 500 to 2000.

The results in Table 1 suggest that BN statistics characterize different model performance. For a better understanding, we randomly sample 20 channels in a residual block and plot the corresponding running statistics of $MBN_{clean}$ and $MBN_{adv}$ in Figure 5. We observe that clean images and adversarial images induce significantly different running statistics, though these images share the same set of convolutional filters for feature extraction. This observation further supports that (1) clean images and adversarial images come from two different domains; and (2) current networks fail to learn a unified representation on these two domains. Interestingly, we also find that adversarial images lead to larger running mean and variance than clean images. This phenomenon is also consistent with the observation that adversarial images produce noisy-patterns/outliers at the feature space (Xie et al., 2019b).

As a side note, this MBN structure is also used as a practical trick for training better generative adversarial networks (GAN) (Goodfellow et al., 2014). Chintala et al. (2016) suggest to construct each mini-batch with only real or generated images when training discriminators, as generated images and real images belong to different domains at an early training stage. However, unlike our situation where BN statistics estimated on different domains remain divergent after training, a successful training of GAN, *i.e.*, able to generate natural images with high quality, usually learns a unified set of BN statistics on real and generated images.

**Training with batch-unrelated normalization layers.** Instead of applying MBN structure to disentangle the mixture distribution, we can also train networks with batch-unrelated normalization layers, which avoids exploiting the batch dimension to calculate statistics, for the same purpose. We choose Group Normalization (GN) for this experiment, as GN can reach a comparable performance to BN on various vision tasks (Wu & He, 2018). Specifically, for each image, GN divides the

| training strategy | clean image accuracy (%) |
|---|---|
| 0% adv + 100% clean | 78.9 |
| MBN$_{clean}$, 100% adv + 100% clean | +0.4 |
| MBN$_{clean}$, 100% adv + 100% clean, ALP | -0.5 |

| training strategy | PGD-2000 accuracy (%) |
|---|---|
| 100% adv + 0% clean | 39.2 |
| MBN$_{adv}$, 100% adv + 100% clean | -0.9 |
| MBN$_{adv}$, 100% adv + 100% clean, ALP | -3.9 |

Table 1: MBN statistics characterize model performance. Using MBN$_{clean}$/MBN$_{adv}$, the trained models achieve strong performance on clean/adversarial images.

| training strategy | PGD-2000 accuracy (%) |
|---|---|
| 100% adv + 0% clean | 39.2 |
| 100% adv + 0% clean* | +3.0 |
| MBN$_{adv}$, 100% adv + 100% clean | 38.3 |
| MBN$_{adv}$, 100% adv + 100% clean* | +1.6 |
| MBN$_{adv}$, 100% adv + 100% clean, ALP | 35.3 |
| MBN$_{adv}$, 100% adv + 100% clean, ALP* | +2.8 |

Table 2: Enforcing a consistent behavior of BN at the training stage and the testing stage significantly boosts adversarial robustness. * denotes that running statistics is used at the last 10 training epochs.

channels into groups and computes the normalization statistics within each group. By replacing all BNs with GNs, the mixture training strategy *100% adv + 100% clean* now can ensure robustness against strong attacks, *i.e.*, the model trained with GN achieves 39.5% accuracy against PGD-500, and increasing attack iterations to 2000 only cause a marginal performance drop by 0.5% (39.0% accuracy against PGD-2000). Exploring other batch-unrelated normalization in adversarial training remains as future work.

**Exceptional cases.** There are some situations where models directly trained with BN can also ensure their robustness against strong attacks, even if clean images are included for adversarial training. Our experiments show constraining the maximum perturbation of each pixel $\epsilon$ to be a smaller value, *e.g.*, $\epsilon = 8$, is one of these exceptional cases. Kannan et al. (2018) and Mosbach et al. (2018) also show that adversarial training with clean images can secure robustness on small datasets, *i.e.*, MNIST, CIFAR-10 and Tiny ImageNet. Intuitively, generating adversarial images on these much simpler datasets or under a smaller perturbation constraint induces a smaller gap between these two domains, therefore making it easier for networks to learn a unified representation on clean and adversarial images. Nonetheless, in this paper, we stick to the standard protocol in Kannan et al. (2018) and Xie et al. (2019b) where adversarial robustness is evaluated on ImageNet with the perturbation constraint $\epsilon = 16$.

### 4.3 REVISITING STATISTICS ESTIMATION OF BN

**Inconsistent behavior of BN.** As the concept of "batch" is not legitimate at inference time, BN behaves differently at training and testing (Ioffe & Szegedy, 2015): during training, the mean and variance are computed on each mini-batch, referred to as *batch statistics*; during testing, there is no actual normalization performed — BN uses the mean and variance pre-computed on the training set (often by running average) to normalize data, referred to as *running statistics*.

For traditional classification tasks, batch statistics usually converge to running statistics by the end of training, thus (practically) making the impact of this inconsistent behavior negligible. Nonetheless, this empirical assumption may not hold in the context of adversarial training. We check this statistics matching of models trained with the strategy *100% adv + 0% clean*, where the robustness against strong attacks is secured. We randomly sample 20 channels in a residual block, and plot the batch statistics computed on two randomly sampled mini-batches, together with the pre-computed running statistics. In Figure 6, interestingly, we observe that batch mean is almost equivalent to running mean, while batch variance does not converge to running variance yet on certain channels. Given this fact, we then study if this inconsistent behavior of BN affects model robustness in adversarial training.

**A heuristic approach.** Instead of developing a new training strategy to make batch statistics converge to running statistics by the end of training, we explore a more heuristic solution: applying pre-computed running statistics for model training during the last 10 epochs. We report the performance comparison in Table 2. By enabling BNs to behave consistently at training and testing, this approach can further boost the model robustness by 3.0% with the training strategy *100% adv + 0% clean*. We also successfully validate the generality of this approach on other two robust training strategies. More specifically, it can improve the model robustness under the training strategies

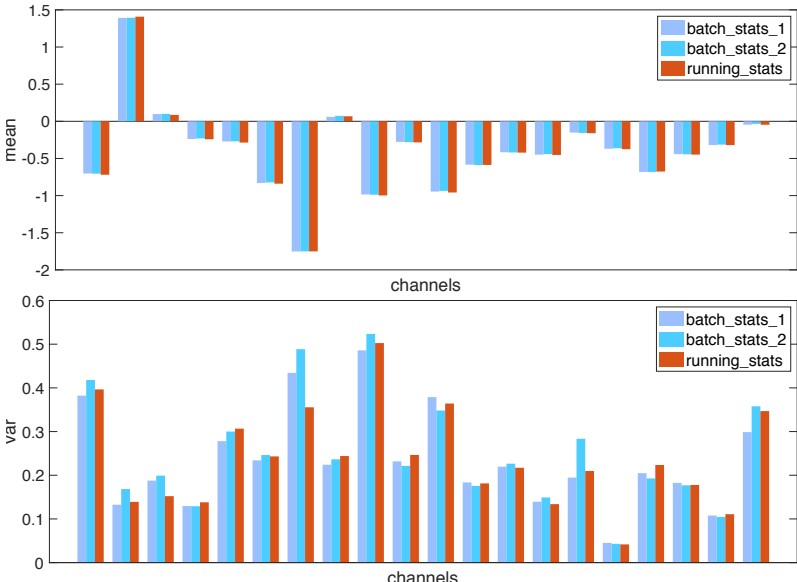

Figure 6: Comparison of batch statistics and running statistics of BN on randomly sampled 20 channels in a ResNet-152's res$_3$ block. We observe that batch mean can converge to running mean, while batch variance still differs from running variance.

$MBN_{adv}$, 100% adv + 100% clean and $MBN_{adv}$, 100% adv + 100% clean, ALP by 1.6% and 2.8%, respectively. These results suggest that model robustness can be benefited from a consistent behavior of BN at training and testing. Moreover, we note this approach does not incur any additional training budgets.

## 4.4 BEYOND ADVERSARIAL ROBUSTNESS

**On the importance of training convolutional filters adversarially.** In Section 4.2, we study the performance of models where the mixture distribution is disentangled for normalization — by applying either $MBN_{clean}$ or $MBN_{adv}$, the trained models achieve strong performance on either clean images or adversarial images. This result suggests that clean and adversarial images share the same convolutional filters to effectively extract features. We further explore whether the filters learned exclusively on adversarial images can extract features effectively on clean images, and vice versa. We first take a model trained with the strategy *100% adv + 0% clean*, and then finetune BNs using only clean images for a few epochs. Interestingly, we find the accuracy on clean images can be significantly boosted from 62.3% to 73%, which is only 5.9% worse than the standard training setting, *i.e.*, 78.9%. These result indicates that convolutional filters learned solely on adversarial images can also be effectively applied to clean images. However, we find the opposite direction does not work — convolutional filters learned on clean images cannot extract features robustly on adversarial images (*e.g.*, 0% accuracy against PGD-2000 after finetuning BNs with adversarial images). This phenomenon indicates the importance of training convolutional filters adversarially, as such learned filters can also extract features from clean images effectively. The findings here also are related to the discussion of robust/non-robustness features in Ilyas et al. (2019). Readers with interests are recommended to refer to this concurrent work for more details.

**Limitation of adversarial training.** We note our adversarially trained models exhibit a performance tradeoff between clean accuracy and robustness — the training strategies that achieve strong model robustness usually result in relatively low accuracy on clean images. For example, *100% adv + 0% clean*, $MBN_{adv}$, *100% adv + 100% clean* and $MBN_{adv}$, *100% adv + 100% clean, ALP* only report 62.3%, 64.4% and 65.9% of clean image accuracy. By replacing BNs with GNs, *100% adv + 100% clean* achieves much better clean image accuracy, *i.e.*, 67.5%, as well maintaining strong robustness. We note that this tradeoff is also observed in the prior work (Tsipras et al., 2018). Besides, Balaji et al. (2019) show it is possible to make adversarially trained models to exhibit a better tradeoff between clean accuracy and robustness. Future attentions are deserved on this direction.

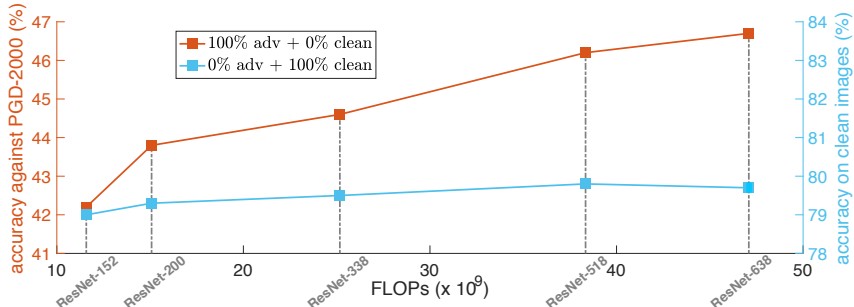

Figure 7: Compared to traditional image classification tasks, adversarial training exhibits a stronger demand on deeper networks. The performance gain of traditional image classification becomes marginal after ResNet-200 while the adversarial robustness continues to grow even for ResNet-638.

## 5 GOING DEEPER IN ADVERSARIAL TRAINING

As discussed in Section 4.2, current networks are not capable of learning a unified representation on clean and adversarial images. It may suggest that the "deep" network we used, *i.e.*, ResNet-152, still underfits the complex distribution of adversarial images, which motivates us to apply larger networks for adversarial training. We simply instantiate the concept of larger networks by going deeper, *i.e.*, adding more residual blocks. For traditional image classification tasks, the benefits brought by adding more layers to "deep" networks is diminishing, *e.g.*, the blue curve in Figure 7 shows that the improvement of clean image accuracy becomes saturated once the network depth goes beyond ResNet-200.

For a better illustration, we train deeper models exclusively on adversarial images and observe a possible underfitting phenomenon as shown in Figure 7. In particular, we apply the heuristic policy in Section 4.3 to mitigate the possible effects brought by BN. We observe that adversarial learning task exhibits a strong "thirst" on deeper networks to obtain stronger robustness. For example, increasing depth from ResNet-152 to ResNet-338 significantly improves the model robustness by 2.4%, while the corresponding improvement in the "clean" training setting (referred to as *0% adv + 100% clean*) is only 0.5%. Moreover, this observation still holds even by pushing the network capacity to an unprecedented scale, *i.e.*, ResNet-638. These results indicate that our so-called "deep" networks (*e.g.*, ResNet-152) are still shallow for the task of adversarial learning, and larger networks should be used for fitting this complex distribution. Besides our findings on network depth, Madry et al. (2018) show increase network width also substantially improve network robustness. These empirical observations also corroborate with the recent theoretical studies (Nakkiran, 2019; Gao et al., 2019) which argues that robust adversarial learning needs much more complex classifiers.

Besides adversarial robustness, we also observe a consistent performance gain on clean image accuracy by increasing network depth (as shown in Table 7). Our deepest network, ResNet-638, achieves an accuracy of 68.7% on clean images, outperforming the relatively shallow network ResNet-152 by 6.1%.

## 6 CONCLUSION

In this paper, we reveal two intriguing properties of adversarial training at scale: (1) conducting normalization in the right manner is essential for training robust models on large-scale datasets like ImageNet; and (2) our so-called "deep" networks are still shallow for the task of adversarial learning. Our discoveries may also be inherently related to our two-domain hypothesis — clean images and adversarial images are drawn from different distributions. We hope these findings can facilitate fellow researchers for better understanding of adversarial training as well as further improvement of adversarial robustness.

**Acknowledgment** This work was supported by ONR N00014-15-1-2356. We would like to thank Kaiming He, Laurens van der Maaten, Judy Hoffman, Yuxin Wu, Yuyin Zhou, Zhishuai Zhang, Yingwei Li and Song Bai for valuable discussions.

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

## A  DIAGNOSIS ON ALP TRAINING PARAMETERS

In the main paper, we note that our reproduced ALP significantly outperforms the results reported in Kannan et al. (2018), as well in an independent study Engstrom et al. (2018). The main differences between our version and the original ALP implementation lie in parameter settings, and are detailed as follows:

- learning rate decay: the original ALP decays the learning rate every two epochs at an exponential rate of 0.94, while ours decays the learning rate by $10\times$ at the 35-th, 70-th and 95-th epoch. To ensure these two policies reach similar learning rates by the end of training, the total number of training epochs of the exponential decay setting and the step-wise decay setting are set as 220 and 110 respectively.
- initial learning rate: the original ALP sets the initial learning rate as 0.045 whereas we set it as 0.1 in our implementation.
- training optimizer: the original ALP uses RMSProp as the optimizer while we use Momentum SGD (M-SGD).
- PGD initialization during training: the original ALP initializes the adversarial perturbation from a random point within the allowed $\epsilon$ cube; while we initialize the adversarial image by its clean counterpart with probability = 0.2, or randomly within the allowed the $\epsilon$ cube with probability = 0.8.
- number of GPUs: the original ALP uses 50 GPUs for adversarial training, while ours uses 128 GPUs.
- network backbone: the original ALP reports results based on Inception-v3 and ResNet-101, while our backbone is ResNet-152.
- PGD-N for training: the original ALP uses PGD-10 for training, while we use PGD-30 for training.

| PGD-N for training | network | #GPUs | PGD initialization | optimizer | initial lr | lr decay | accuracy (%) | |
|---|---|---|---|---|---|---|---|---|
| | | | | | | | PGD-10 | PGD-2000 |
| 10 | ResNet-101 | 48 | P($\epsilon$-cube) = 1.0 | RMSProp | 0.045 | exponential | 38.1 | 2.1 |
| | | | | | | step-wise | 47.0 | 2.7 |
| | | | | | | | 50.2 | 0.8 |
| | | | | | | | 48.2 | 0.7 |
| | | | P($\epsilon$-cube) = 0.8 P(clean) = 0.2 | M-SGD | 0.1 | | 47.0 | 2.1 |
| | | 128 | | | | | 47.5 | 2.9 |
| | | | | | | | 47.3 | 3.3 |
| 30 | ResNet-152 | | | | | | 48.8 | 23.0 |

Table 3: The results of ALP re-implementations under different parameter settings. We show that applying stronger attackers for training, *e.g.*, change from PGD-10 to PGD-30, is the most important factor for achieving strong robustness. Other parameters, like optimizer, do not lead to significant robustness changes.

By following the parameter settings listed in the ALP paper[4], we can train a ResNet-101 with an accuracy of 38.1% against PGD-10. The ResNet-101 performance reported in the ALP paper is 30.2% accuracy against an attack suite[5]. This ~8% performance gap is possibly due to different attacker settings in evaluation. However, by evaluating this model against PGD-2000, we are able to obtain a similar result that reported in Engstrom et al. (2018), *i.e.*, Engstrom et al. (2018) reports ALP obtains 0% accuracy, and in our implementation the accuracy is 2.1%.

Given these different settings, we change them one by one to train corresponding models adversarially. The results are summarized in Table 3. Surprisingly, we find the most important factor for the performance gap between original ALP paper and ours is the attacker strength used for training — by changing the attacker from PGD-10 to PGD-30 for training, the robustness against PGD-2000 can be increased by 19.7%. Other parameters, like network backbones or the GPU number, do not lead to significant performance changes.

---

[4]For easier implementation, we apply 48 GPUs (which can be distributed over 6 8-GPU machines) for adversarial training, instead of using the original number, *i.e.*, 50 GPUs.

[5]This attack suite contains 8 different attackers, including PGD-10. However, due to the vague description of parameter settings in this attack suite, we are not able to reproduce it.

# B EXPLORING THE IMPACT OF PARAMETER SETTINGS IN ADVERSARIAL TRAINING

In this section, we explore the impact of different training parameters on model performance.

## B.1 PGD-N FOR TRAINING

As suggested in Table 3, the number of attack iteration used for training is an important factor for model robustness. We hereby provide a detailed diagnosis of model performance trained with PGD-$\{5, 10, 20\}$[6] for different training strategies. We report the performance in Table 4, and observe that decreasing the number of PGD attack iteration used for training usually leads to weaker robustness. Nonetheless, we note the amount of this performance change is strongly related to training strategies. For strategies that cannot lead to models with strong robustness, *i.e.*, *100% adv + 100% clean* and *100% adv + 100% clean, ALP*, this robustness degradation is extremely severe (which is similar to the observation in Table 3). For example, by training with PGD-5, these two strategies obtains nearly no robustness, *i.e.*, ~0% accuracy against PGD-2000. While for strategies that can secure model robustness against strong attacks, changing from PGD-30 to PGD-5 for training only lead to a marginal robustness drop.

| PGD-N for training | accuracy against PGD-2000 (%) | | | | |
|---|---|---|---|---|---|
| | 100% adv + 0% clean | 100% adv + 100% clean | 100% adv + 100% clean, ALP | MBN$_{adv}$, 100% adv + 100% clean | MBN$_{adv}$, 100% adv + 100% clean, ALP |
| 30 | 39.2 | 20.9 | 23.0 | 38.3 | 35.3 |
| 20 | -1.0 | -7.3 | -3.8 | -4.2 | +0.5 |
| 10 | -2.4 | -17.7 | -19.7 | -5.5 | -1.4 |
| 5 | -3.3 | -20.9 | -22.7 | -7.1 | -2.4 |

Table 4: **Robustness evaluation of models adversarially trained with PGD-$\{30, 20, 10, 5\}$ attackers**. We observe that decreasing the number of PGD attack iteration for training usually leads to weaker robustness, while the amount of degraded robustness is strongly related to training strategies.

## B.2 APPLYING RUNNING STATISTICS IN TRAINING

In Section 4.3 (of the main paper), we study the effectiveness of applying running statistics in training. We hereby test this heuristic policy under more different settings. Specifically, we consider 3 strategies, each trained with 4 different attackers (*i.e.*, PGD-$\{5, 10, 20, 30\}$), which results in 12 different settings. We report the result in Table 5. We observe this heuristic policy can boost robustness on *all* settings, which further supports the importance of enforcing BN to behave consistently at training and testing.

| training strategy | accuracy against PGD-2000 (%) | | | |
|---|---|---|---|---|
| | PGD-5 | PGD-10 | PGD-20 | PGD-30 |
| 100% adv + 0% clean | 35.9 | 36.8 | 38.2 | 39.2 |
| 100% adv + 0% clean* | +1.9 | +2.4 | +1.8 | +3.0 |
| MBN, 100% adv + 0% clean | 31.2 | 32.8 | 34.1 | 38.3 |
| MBN, 100% adv + 0% clean* | +3.0 | +3.3 | +2.4 | +1.6 |
| MBN, 100% adv + 0% clean, ALP | 32.9 | 33.9 | 35.8 | 35.3 |
| MBN, 100% adv + 0% clean, ALP* | +2.5 | +2.8 | +1.3 | +2.8 |

Table 5: Validating the effectiveness of applying running statistics in training on more settings. We observe this heuristic policy can boost robustness on all settings. * denotes that running statistics is used at the last 10 training epochs.

| training batch size | accuracy against PGD-2000 (%) |
|---|---|
| 4096 | 42.2 |
| 512 | -0.2 |
| 1024 | +0.9 |
| 2048 | +1.1 |
| 8192 | -0.3 |

Table 6: Performance evaluation of models trained with different batch size. The best performance can be achieved by training with a batch size of 2048.

## B.3 BATCH SIZE IN TRAINING

The default number of training batch size is 4096. We hereby study the model performance when training with the batch size of $\{512, 1024, 2048, 8192\}$, respectively. Without loss of generality,

---

[6]For PGD-5 and PGD-10, we set the attack step size $\alpha$ to be 4 and 2, respectively.

we study the training strategy *100% adv + 0% clean*. The heuristic policy in Section 4.3 (of the main paper) is applied to achieve stronger robustness. Compared to the default setting (*i.e.*, 4096 images/batch), training with smaller batch size leads to better robustness. For example, changing batch size from 4096 to 1024 or 2048 can improve the model robustness by ~1%. While training with much smaller (*i.e.*, 512 images/batch) or much larger (*i.e.*, 8192 images/batch) batch size results in a slight performance degradation.

## C  PERFORMANCE OF ADVERSARIALLY TRAINED MODELS

In the main paper, our study is driven by improving adversarial robustness (measured by the accuracy against PGD-2000), while leaving the performance on clean images ignored. For the completeness of performance evaluation, we list the clean image performance of these adversarially trained models in Table 7. Moreover, to facilitate performance comparison in future works, we list the corresponding accuracy against PGD-{10, 20, 100, 500} in this table as well.

| network | training batch size | PGD-N for training | setting | accuracy (%) | | | | | |
|---|---|---|---|---|---|---|---|---|---|
| | | | | clean | PGD | | | | |
| | | | | | 10 | 20 | 100 | 500 | 2000 |
| ResNet-152 | 4096 | 30 | 100% adv + 100% clean, ALP | 73.5 | 48.8 | 46.3 | 34.1 | 27.3 | 23.0 |
| | | | 100% adv + 100% clean | 78.0 | 56.2 | 53.0 | 35.6 | 26.6 | 20.9 |
| | | | 100% adv + 80% clean | 77.4 | 56.8 | 54.6 | 39.6 | 28.7 | 20.3 |
| | | | 100% adv + 60% clean | 75.5 | 54.6 | 52.1 | 40.3 | 23.5 | 15.3 |
| | | | 100% adv + 40% clean | 73.9 | 54.1 | 51.5 | 40.2 | 33.6 | 29.1 |
| | | | 100% adv + 20% clean | 68.0 | 54.6 | 51.5 | 40.6 | 34.6 | 32.1 |
| | | | 100% adv + 0% clean | 62.3 | 52.5 | 50.0 | 41.7 | 39.6 | 39.2 |
| | | | 100% adv + 0% clean* | 62.1 | 52.4 | 50.3 | 43.9 | 42.6 | 42.2 |
| | | | MBN$_{adv}$, 100% adv + 100% clean | 64.4 | 51.8 | 49.1 | 40.9 | 38.8 | 38.3 |
| | | | MBN$_{adv}$, 100% adv + 100% clean* | 64.2 | 52.5 | 50.0 | 42.1 | 40.5 | 39.9 |
| | | | MBN$_{adv}$, 100% adv + 100% clean, ALP | 65.9 | 47.3 | 45.0 | 38.3 | 35.9 | 35.3 |
| | | | MBN$_{adv}$, 100% adv + 100% clean, ALP* | 64.3 | 49.0 | 47.2 | 40.4 | 38.6 | 38.1 |
| | | | GN, 100% adv + 100% clean | 67.5 | 52.1 | 49.3 | 41.9 | 39.5 | 39.0 |
| | | 20 | 100% adv + 100% clean, ALP | 72.7 | 50.1 | 47.6 | 33.4 | 25.2 | 19.2 |
| | | | 100% adv + 100% clean | 78.4 | 55.9 | 52.5 | 31.7 | 20.0 | 13.6 |
| | | | 100% adv + 0% clean | 62.5 | 54.1 | 51.1 | 41.8 | 39.0 | 38.2 |
| | | | 100% adv + 0% clean* | 65.4 | 53.9 | 51.0 | 42.7 | 40.8 | 40.0 |
| | | | MBN$_{adv}$, 100% adv + 100% clean | 67.9 | 52.8 | 49.9 | 39.3 | 35.5 | 34.1 |
| | | | MBN$_{adv}$, 100% adv + 100% clean* | 67.2 | 52.7 | 49.9 | 40.2 | 37.2 | 36.5 |
| | | | MBN$_{adv}$, 100% adv + 100% clean, ALP | 68.1 | 48.8 | 46.7 | 39.5 | 36.7 | 35.8 |
| | | | MBN$_{adv}$, 100% adv + 100% clean, ALP* | 67.4 | 50.0 | 47.7 | 40.3 | 37.4 | 37.1 |
| | | 10 | 100% adv + 100% clean, ALP | 74.9 | 47.3 | 47.1 | 22.4 | 7.3 | 3.3 |
| | | | 100% adv + 100% clean | 78.4 | 56.6 | 55.1 | 28.8 | 7.1 | 3.2 |
| | | | 100% adv + 0% clean | 66.0 | 53.6 | 50.9 | 41.1 | 38.0 | 36.8 |
| | | | 100% adv + 0% clean* | 65.9 | 53.5 | 50.7 | 42.3 | 39.9 | 39.2 |
| | | | MBN$_{adv}$, 100% adv + 100% clean | 68.7 | 53.5 | 49.9 | 38.7 | 34.3 | 32.8 |
| | | | MBN$_{adv}$, 100% adv + 100% clean* | 67.8 | 52.6 | 49.8 | 40.0 | 36.9 | 36.1 |
| | | | MBN$_{adv}$, 100% adv + 100% clean, ALP | 68.7 | 49.0 | 46.8 | 38.9 | 35.4 | 33.9 |
| | | | MBN$_{adv}$, 100% adv + 100% clean, ALP* | 67.7 | 50.0 | 47.7 | 40.2 | 37.4 | 36.7 |
| | | 5 | 100% adv + 100% clean, ALP | 75.0 | 49.6 | 37.6 | 5.7 | 0.8 | 0.3 |
| | | | 100% adv + 100% clean | 78.6 | 41.5 | 17.6 | 0.2 | 0.0 | 0.0 |
| | | | 100% adv + 0% clean | 67.0 | 54.0 | 50.5 | 40.7 | 37.5 | 35.9 |
| | | | 100% adv + 0% clean* | 66.9 | 53.6 | 50.4 | 41.5 | 38.8 | 37.8 |
| | | | MBN$_{adv}$, 100% adv + 100% clean | 69.3 | 53.0 | 49.8 | 38.0 | 33.3 | 31.2 |
| | | | MBN$_{adv}$, 100% adv + 100% clean* | 68.9 | 53.1 | 49.8 | 39.8 | 35.9 | 34.2 |
| | | | MBN$_{adv}$, 100% adv + 100% clean, ALP | 69.2 | 49.6 | 46.8 | 38.9 | 34.9 | 32.9 |
| | | | MBN$_{adv}$, 100% adv + 100% clean, ALP* | 69.2 | 50.4 | 48.3 | 40.3 | 37.0 | 35.4 |
| | 512 | 30 | 100% adv + 0% clean* | 62.6 | 52.4 | 50.4 | 43.8 | 42.4 | 42.0 |
| | 1024 | | | 63.1 | 53.3 | 51.0 | 44.6 | 43.5 | 43.1 |
| | 2048 | | | 62.7 | 53.3 | 50.7 | 45.0 | 43.7 | 43.3 |
| | 8196 | | | 61.8 | 52.1 | 49.8 | 43.4 | 42.3 | 41.9 |
| ResNet-200 | 4096 | | | 63.5 | 53.8 | 51.4 | 45.1 | 43.9 | 43.8 |
| ResNet-338 | | | | 65.4 | 55.5 | 53.2 | 46.6 | 45.1 | 44.6 |
| ResNet-518 | | | | 66.7 | 56.7 | 54.4 | 48.1 | 46.4 | 46.2 |
| ResNet-638 | | | | 67.2 | 57.2 | 54.8 | 48.7 | 47.1 | 46.7 |

Table 7: For easier benchmarking in future works, we list the detailed performance of adversarially trained models. * denotes running statistics is used at the last 10 training epochs.

