# OpenReview forum: "Intriguing Properties of Adversarial Training at Scale"
_ICLR.cc/2020/Conference — Accept (Poster)_

### Official Review · AnonReviewer1 · 2019-10-24
**Official Blind Review #1**

**Rating:** 6

**Review:**

This paper reveals some interesting properties of neural networks when trained adversarially at ImageNet scale. The total cost of the experiments is quite impressive, therefore the results are valuable references. With extensive experiments, the authors reveals two intriguing properties of neural networks when trained adversarially, and devotes most of the paper to studying the first one, i.e., why networks with Batch Normalization cannot achieve high robustness when both clean and adversarial images are used for training. I am not fully convinced by this point, but I do think the second point is very interesting. Different from previous work arguing that more data is needed for improving adversarially robust generalization, this paper shows that adversarial robustness can be improved consistently just by making ResNet deeper.

I tend to accept this paper for the valuable results and the discovery of the positive correlation between network capacity and adversarial robustness, but I am not fully convinced by the explanations for the problems with BN. I hope the authors could address my concerns before I can be more confident about my decision.

To me, it seems the correct way to do adversarial training is to only use adversarial samples, since we are trying to minimize the maximum risk inside a norm ball around the clean sample (Madry et al. 2018). All the experiments with clean images in the objective are consistently worse than training only on adversarial samples under the same setting in this paper. It therefore becomes not that important to study the effect of Batch Normalization on training with both clean and adversarial images.

Also, I am a little bit unconvinced that the running mean and variance of BN is the cause of bad performance for the mixed training. First, I want to know the standard and adversarial accuracies of the network with GN in the "100% adv + 0% clean" setting. It seems missing in the paper. Despite being much better than BN in the "100% adv + 100% clean" setting, it is not sure whether it is caused by the improvement in some other property (e.g., capacity) from GN. If the robust accuracy with "100% adv + 0% clean" is also much higher with GN than BN (seems unlikely though), then replacing GN with BN does not solve the problem and it is still the objective to blame.

I am also not fully convinced by the experiments with separated BN parameters for clean and adversarial samples. By doing so, the network is actually trained to approximate its behavior in the "100% adv + 0% clean" setting. Since the adversary is maximizing the loss, the gradient (of the conv layers) from adversarial samples will dominate the total gradient in the "MBN 100% adv + 100% clean" setting, and the "MBN_{adv}" network will be trained similarly as the network trained in the "100% adv + 0% clean" setting. This explains why the adversarial accuracy can be very close to "100% adv + 0% clean" but still lower.  Comparing results (under different ratios of clean samples) for networks without BN is very important and much more effective at explaining the phenomenon than trying to make BN work.

It would also be great if the authors could provide some curve showing the tendency of the running variance of BN. Sec 4.3 does make lots of sense from a practical aspect, i.e., fixing mean and var for training in the last 10 epochs could improve the results using same number of epochs, but what if we just train more epochs? Will the variance converge in just tens of epochs?

Finally, though it seems that deeper networks are more robust,  the robustness might be a misconception caused by gradient vanishing. Could the authors provide the average gradient norms on the correctly-classified images (remaining correct after attack) in the first step of PGD attacks for the models in Figure 7? If deeper networks indeed have much smaller gradient norm, could the authors try scaling the loss by some factor to make the attacks stronger?


===========================================================
I am not fully convinced by your explanations. Could you give the results for the gradients, or use the loss function from CW attack + PGD to report the robustness of the deeper models? I am not sure which type of residual connection you are using, but if you are using the "original" version as mentioned in this paper [1], then it does not necessarily avoid gradient vanishing, since if ReLU is deactivated, the gradient will be brought to zero. Another possible cause for the possible misconception of robustness might be caused by the saturated cross entropy loss, which could also give 0 gradients but could be verified by switching to the margin loss as in the CW attack. I have encountered such cases when I achieved non-zero robust accuracy (evaluated with PGD on cross entropy loss, and the robust accuracy is similar to your improvement from ResNet152 to ResNet638) on a naturally trained network as I increased the number of layers, but when I switched to CW loss (or margin loss), the robust accuracy goes to zero. Even for resnet, when you multiply the number of layers by ~4 as done in the paper, such phenomenon will actually happen.

[1] He, Kaiming, et al. "Identity mappings in deep residual networks." European conference on computer vision. Springer, Cham, 2016.

=====================================================================
Raising my score. Found a recent good paper based on the findings of this paper.

**Experience Assessment:**

I have published one or two papers in this area.

**Review Assessment: Checking Correctness Of Derivations And Theory:**

N/A

**Review Assessment: Checking Correctness Of Experiments:**

I carefully checked the experiments.

**Review Assessment: Thoroughness In Paper Reading:**

I made a quick assessment of this paper.

---

> ### Author Response · Authors · 2019-11-15
> **Responses to Reviewer #1**
>
> We first thank the reviewer for the detailed comments and the appreciation of our analysis on network depth. For the concerns about BN in adversarial training, we address as follows:
>
> C1: Is adversarial training with clean images meaningful?
> A1: We argue that adversarial training with clean images is meaningful. As evaluation on clean images is the main metric for assessing model performance in the current computer vision research field, it is reasonable to always ask models to train with clean images (in addition to adversarial examples); meanwhile,  training exclusively on adversarial examples results in significant performance drop on clean images accuracy, which hampers the usage of these robust models in practice. Given these facts, a lot of works have attempted to make networks have both good clean image accuracy and strong robustness by training with a mixture of clean and adversarial images.
>
> However, previous works show that training with a mixture of clean and adversarial images generally cannot have strong robustness. Motivated by this observation, we hereby provide a detailed study on this research direction and identify (1) adversarial and clean images are drawn from two domains; and (2) the devil lies in BN. We believe our findings are useful for future research in this direction, e.g., to obtain both strong robustness and good clean image accuracy.
>
> C2: GN results?
> A2: By training GNs with “100% adv + 0% clean”, it achieves 56.9% accuracy on clean images, and 40.6% accuracy against PGD-2000. Compared to BNs, GNs achieve lower clean image accuracy (-5.4%), and slightly better on robustness (+1.4%). Since the accuracy and the adversarial robustness exhibit a trade-off [1] (Section 4.4), we do not think replacing BNs with GNs brings extra capacity to networks given this result. Therefore, we believe our analysis in the paper is still valid: mitigating the confusion of normalization statistics estimation w.r.t. a mixture distribution is crucial for achieving strong robustness.
>
> C3: Separate BNs?
> A3: We keep separate BNs to guarantee the normalization process is conducted exclusively only on a single domain. While for all other layers (e.g., conv layers and fc layers), they are jointly optimized over both adversarial images and clean images. This training manner is still different from training exclusively either on clean images or adversarial examples. We believe such trained convolutional layers possess both adversarial features and clean images, thus may show better generalization on other tasks, like against common corruption on ImageNet-C.
>
> Moreover, these experiments on separate BNs suggest that carefully handling normalization is crucial for network robustness, and provide an explanation on why previous mixture adversarial training cannot reach strong robustness.
>
> C4: Training with more epochs?
> A4: Interesting, our preliminary results show that training with more epochs provides an opposite result---networks get better clean image accuracy but worse adversarial robustness. We plan to take a deep look at this phenomenon in the future.
>
> C5: Robustness might be a misconception caused by gradient vanishing?
> A5: All our experiments are conducted on ResNet [2], which successfully utilizes residual learning to address the optimization difficulties even for VERY DEEP networks. Besides, for the ResNet structure, every building block has a residual connection. By completely go through this residual path, even with the very deep Res-638, there only has four convolutional layers, which are used for reducing the spatial dimension of feature maps, between the input layer and the output layer.
>
> Intuitively, if vanishing gradients happen here, networks should have (1) higher training error as they cannot optimize the network well; or (2) similar clean image accuracy but stronger robustness as vanished gradients confuse the attacker. However, in our ResNet-638 trained with “100% adv + 0% clean” setting, it shows (1) much lower training error than shallower networks; and (2) stronger performance on both clean image accuracy and adversarial robustness.
>
> Therefore, we do not think the gradient vanishing problem happens in our very deep network experiments, and our conclusion remains valid.
>
> [1] Hongyang Zhang, Yaodong Yu, Jiantao Jiao, Eric P Xing, Laurent El Ghaoui, and Michael I Jordan. Theoretically principled trade-off between robustness and accuracy. In ICML, 2019b.
> [2] Kaiming He, Xiangyu Zhang, Shaoqing Ren, and Jian Sun. Deep residual learning for image recognition. In CVPR, 2016.

---

> > ### Author Response · Authors · 2019-12-20
> > **LATE Responses to Reviewer #1**
> >
> > ###########################################################
> > Thanks for the detailed comments. We are sorry for this late response, as we are not allowed to post anything after Nov 15.  We hope the following preliminary results can convince the reviewer that applying adversarial training with deeper networks indeed improves robustness.
> >
> > Due to resource and time limitation, all the results are reported using Res338 & Res638 on randomly selected 10000 ImageNet images. Following the reviewer’s suggestions, we conduct two sets of experiments to exclude the possibility of gradient vanishing during attacks: (1) enhance attacker strength by scaling the cross entropy loss by 10 or 100; or (2) instead of using cross entropy loss, we directly maximize the logits of the target class for attacking (similar to CW attacks). The results are reported as follows:
> >
> > Part 1 --- Scale cross entropy loss during attacks
> >
> > | Model    | Scaling Loss  | PGD-2000 accuracy (%)  |
> > | ----------- | ------------------  | --------------------------------- |
> > | Res338  |        1X            |                   42.7                |
> > | Res638  |        1X            |                47.2 (+4.5)        |
> > | ----------- | ------------------  | --------------------------------- |
> > | Res338  |        10X          |                   42.5                |
> > | Res638  |        10X          |                47.2 (+4.3)        |
> > | ----------- | ------------------  | --------------------------------- |
> > | Res338  |        100X        |                   42.5                |
> > | Res638  |        100X        |                47.2 (+4.3)        |
> >
> > From the table above, we can see under all situations, Res638 consistently demonstrates better robustness than Res338. Besides, we note that scaling loss by some factors does not hurt model performance more when evaluated at 2000-step PGD attacker. It is possible due to that PGD-2000 is already a very strong attack, which provides a thoroughly evaluation on model robustness (but we indeed see that this scaling loss strategy leads to worse accuracy if evaluated on some weaker attackers like PGD-20).
> >
> > Part 2 ---  Directly maximize the logits of the target class (without using cross entropy loss)
> > | Model    | PGD-2000 accuracy (%)  |
> > | ----------- | --------------------------------- |
> > | Res338  |                   50.3                |
> > | Res638  |               53.3 (+3.0)         |
> >
> > The cross entropy loss term is removed here and we directly maximize the logits of the target class during attacks. From the table above, we observe that Res638 still performs better than Res338.
> >
> > All the results above corroborate our robustness evaluation in the main paper is correct and is not led by the misconception of gradient vanishing. We hope these results can address the reviewer’s concerns.

---

### Official Review · AnonReviewer2 · 2019-10-26
**Official Blind Review #2**

**Rating:** 8

**Review:**

The paper studies adversarial training "at scale", i.e., on the ImageNet dataset. The paper makes two main contributions in this context:
- An in-depth investigation of the effect batch normalization (BN) has on adversarial robustness when the network is trained adversarially.
- Training increasingly deeper residual networks (up to 600 layers) and demonstrating that adversarial robustness still increases in this regime (unlike standard accuracy).

Overall I find the findings presented in the paper interesting and recommend accepting the paper. Experimenting with adversarial training on ImageNet is still hard for many academic groups due to the high computational cost. Hence the results of the paper may be useful for the wider robustness community. To achieve this goal, I strongly encourage the authors to release their models in a format that is easy to build on and experiment with for other researchers (e.g., PyTorch model checkpoints). Moreover, I find it interesting that very deep models on ImageNet can achieve increased adversarial robustness. To the best of my knowledge, these are the best robustness numbers published on ImageNet.

Further comments and questions:

- It would be good to know if BN also affects adversarial robustness on CIFAR-10 or other datasets.

- What happens when the width of the network is increased? Does this also help adversarial robustness?

- Section 4.1 states that "Adversarial training can be dated back to (Goodfellow et al., 2015), [...]". While the specific form of adversarial training for adversarial robustness in CNNs is indeed recent, it may be helpful for readers to provide additional context, e.g., min-max formulations have a long history in robust optimization and statistics.

- Section 4.4 states "Interestingly, we find that the accuracy on clean images can be significantly boosted from 62.3% to 73%.". It would be good to add context and state what accuracy the network achieves with standard training.

**Experience Assessment:**

I have published in this field for several years.

**Review Assessment: Checking Correctness Of Derivations And Theory:**

N/A

**Review Assessment: Checking Correctness Of Experiments:**

I assessed the sensibility of the experiments.

**Review Assessment: Thoroughness In Paper Reading:**

I read the paper at least twice and used my best judgement in assessing the paper.

---

> ### Author Response · Authors · 2019-11-15
> **Responses to Reviewer #2**
>
> We first thank the reviewer for the detailed comments and the appreciation of our work. We will try our best to make these large models be easily accessed from academic groups. We address the concerns below:
>
> Q1: BN on CIFAR-10?
> A1: The discussion of BN on CIFAR-10 can be found in the last paragraph of Section 4.2. For adversarially trained models on small datasets, like MNIST, CIFAR-10 and Tiny-ImageNet, BN will NOT cause problems on model robustness. Our argument is that these dataset are relatively easier for learning, and many existing architectures already have enough capacity to learn a unified representation on both clean and adversarial images, therefore ease the confusion of statistics estimation at BN. While this paper stands on the perspective of adversarial training at SCALE, which reveals some generally important & principle issues which are not observable on small datasets.
>
> Q2: Width of networks?
> A2: [1] has discussed the relationship between network width and adversarial robustness---adversarially trained wider networks exhibit stronger robustness. As suggested by R3, we have added this related finding in Section 5 for a better view to future readers.
>
> In general, we hold the view that a better adversarial training requires larger networks (either in terms of width or depth or both).
>
> Q3: Related works in min-max formation.
> A3: Thanks for this great suggestion! To our best knowledge, min-max type of formulation can be dated as early as to [2] and we have included in the updated version accordingly. Please note if there are other references to be discussed and cited.
>
> Q4: Statement confusion in Sec 4.4.
> A4: Thanks for this suggestion! The standard training setting achieve 78.9% accuracy. We have updated the paper accordingly.
>
> [1] Madry, Aleksander, et al. "Towards deep learning models resistant to adversarial attacks." ICLR 2018.
> [2] Abraham Wald. Statistical decision functions which minimize the maximum risk. In Annals of Mathematics, 1945.

---

### Official Review · AnonReviewer3 · 2019-10-28
**Official Blind Review #3**

**Rating:** 6

**Review:**

This paper introduces two properties of adversarial training observed from abundant empirical results. Based on the discoveries, the authors propose plausible explanations as well as new methods to gain higher adversarial robustness. The two properties are as follows

1. The batch normalization may negatively affect the adversarial robustness, where a training batch consists of a mixture of clean and adversarial examples. The authors observe that the parameters for batch normalization (BN) may be quite different between batches of clean and of adversarial examples, and conjecture the reason as that these two sets of examples are from two different domains. Therefore, the authors propose a few methods to boost the adversarial robustness: using different BN stats for clean and adversarial examples (but keeping the other parameters shared, which may seem useful only for fundamental study due to the requirement of clean/adversarial label for each example), using batch-unrelated normalization, and changing stat estimation for BN to reduce the difference between training and inference steps.

2. The deep networks for general image classification may be too shallow for adversarial training. Since the mixture of clean and adversarial examples may form a two domain distribution that is challenging to model, typical neural networks (even for ResNet-152 that has high depth for general image classification) may have too low capacity. The authors show that increasing the depth of neural network (up to and possibly beyond ResNet-638) results in even higher accuracy.

To my best understanding, the results and analysis of this paper are valid, and the proposed methods have shown gains in abundant experiments. Due to the significance of adversarial training and the new discoveries of this paper, I think the contributions are sufficient and would lean towards the paper being published / weak accept. However, below are my questions and concerns that I would like the authors to address.

1. For this paper, the adversarial examples are generated from a particular class of attacker, namely Projected Gradient Descent. I am curious how the conclusion could generalize and help the robustness if we have more than one adversarial attackers.

2. The paper focuses on adversarial robustness and seems to deprioritize clean image accuracy, which could seem to limit the scope for application purposes. Frankly, I agree that a fundamental understanding of adversarial robustness would be significant, and the authors discuss the problem of relatively lower clean image accuracy in Section 4.4. However, I might feel that clean image accuracy should be as significant for practical purposes, and it would be great if we can balance the trade-off between clean image accuracy and the adversarial robustness. The authors should feel free to correct me if I do not understand correctly.

3. There seems to be an interesting observation in Fig 1 for which I am curious. The accuracy for PGD-2000 in Fig 1 does not go monotonically with the ratio of clean images — the lowest accuracy is at 60 percent of clean images, which seems not to fully align with the argument that removing clean images will help robustness. Personally it would be great if the authors could share their insights of possible reasons.

4. For the second discovery that deeper networks help adversarial robustness, the red line (for adversarial robustness) in Fig 7 seems not converged yet at ResNet-638. If the computation resources allow, I am curious on the depth of ResNet at which the red line becomes flat, and this could be useful for headroom analysis on how good accuracy we can reach.

5. For the second discovery that deeper networks help adversarial robustness (Section 5), it seems Madry et al 2018 (Towards deep learning models resistant to adversarial attacks) also discusses model capacity vs the adversarial robustness. The mentioned paper does not seem to use deeper structure but uses other ways to increase capacity. The mentioned paper has been referred to in other sections of this work, however, it may be good to contrast in Section 5 on the conceptual novelty in this paper.

6. Typos: The title of Table 1, “MBN_{clean}/MBN_{clean}” would be “MBN_{clean}/MBN_{adv}”. There are some “, ,” (double commas) in the appendices. Comma after ‘(’ in appendix B3. Overall this paper is well written and easy to follow.

**Experience Assessment:**

I have read many papers in this area.

**Review Assessment: Checking Correctness Of Derivations And Theory:**

N/A

**Review Assessment: Checking Correctness Of Experiments:**

I carefully checked the experiments.

**Review Assessment: Thoroughness In Paper Reading:**

I read the paper thoroughly.

---

> ### Author Response · Authors · 2019-11-15
> **Responses to Reviewer #3**
>
> We first thank the reviewer for the detailed comments and the appreciation of our work. We address the concerns below:
>
> Q1: Generalize to more adversarial attacks?
> A: As suggested in [1],  PGD is a UNIVERSAL first order adversary. Many works [2,3] later corroborate that training with PGD alone already secure strong robustness against other attacks, which suggests that PGD may provide a good description of the adversarial distribution. Therefore, we strongly believe our results on PGD can generalize to other adversarial attacks.
>
> If we have more than one attackers for training, these attackers then together can provide a more holistic description of the distribution of adversarial examples, therefore should enable models to improve robustness further.
>
> Q2: Accuracy drop of adversarial training?
> A: Yes, we strongly agree with the reviewer that clean image accuracy is VERY important. Our ultimate goal is to find a model with a better tradeoff. However, we do not think it is achievable without a fundamental understanding of adversarial robustness. Motivated by this, we hereby present a detailed analysis and reveal several intriguing properties of adversarial training. We sincerely hope our study can benefit future research in this direction.
>
> Q3: Curve in Fig.1 is not monotonical?
> A: We believe the main reason for this observation is that the estimation of the statistics at BN is not correct. With this biased normalization process, the training can be unstable and leads to this “outlier” data point in Fig. 1. Though the overall process is not strictly monotonical, we can see that gradually removing clean images from the training set improves model robustness in general.
>
> Q4: Beyond ResNet-638?
> A: We are sorry that we do not have enough resources for training deeper networks during the rebuttal period. But our Fig.7 still delivers an exciting signal---it is possible to reach higher robustness by keeping training with deeper networks, and supports our argument that current deep networks are still “shallow” for the task of adversarial training.  A detailed study of the relationship between network capacity (e.g., depth, width) and robustness will be provided in our future work.
>
> Q5&6
> A: Thanks! We have updated the paper accordingly.
>
> [1] Madry, Aleksander, et al. "Towards deep learning models resistant to adversarial attacks." ICLR 2018.
> [2] Athalye, Anish, et al. "Obfuscated gradients give a false sense of security: Circumventing defenses to adversarial examples." ICML 2018.
> [3] Mosbach, Marius, et al. "Logit pairing methods can fool gradient-based attacks." arXiv 2018.

---

### Author Response · Authors · 2019-11-15
**A General Response & A Change in Paper Contribution**

We fist thank all reviewers for their valuable comments, which help us to improve the quality of this paper. We are glad to see that all reviewers reach the consensus for accepting this paper, and acknowledge the importance of this paper on studying adversarial robustness.

Prior to our responses, we would like to indicate there is a slight contribution change in our paper. In Section 4.2, we present two ways for disentangling the mixture distribution during adversarial training: (1) mixture BN and (2) GN. In our updated paper version, instead of claiming “we PROPOSE MBN for xxx”, we soften it to “we APPLY MBN for xxx”, and cite MBN as a contribution proposed in [1].

We first want to highlight that our paper has significant differences to [1]: [1] is driven by clean image accuracy, without discussing anything related to adversarial robustness. While this paper mainly focuses on adversarial robustness, and reveal several intriguing properties in adversarial training for securing/enhancing model robustness.

Though we change the statement on the originality of MBN, the main contributions of this paper stay the same:
(1) We provide a set of ablations to show that traditional BN may negatively impact model robustness if trained with a mixture of clean and adversarial images;
(2) We demonstrate disentangling the mixture distribution at normalization layers (by using either MBN or GN) can successfully secure model robustness;
(3) We propose a heuristic approach to properly handling BN statistics for improving adversarial robustness further;
(4) We discuss the relationship between network depth and model robustness, and suggest that current deep networks may be still “shallow” for the task of adversarial learning;

Our detailed responses to each reviewer can be found below.


[1] Anonymous. Adversarial examples improve image recognition. In Submission, 2019.

---

### Decision · Program_Chairs · 2019-12-19

**Decision:**

Accept (Poster)

**Comment:**

This paper studies the properties of adversarial training in the large scale setting. The reviewers found the properties identified by the paper to be of interest to the ICLR community - in particular the robustness community. We encourage the authors to release their models to help jumpstart future work building on this study.